Genome-wide characterization and expression analysis of aquaporins in salt cress (Eutrema salsugineum)

Qian Weiguo
Yang Xiaomin
Li Jiawen
Luo Rui
Yan Xiufeng
Pang Qiuying qiuying@nefu.edu.cn
Alkali Soil Natural Environmental Science Center, Northeast Forestry University/Key Laboratory of Saline-alkali Vegetation Ecology Restoration in Oil Field, Ministry of Education , Harbin , China
Balao Francisco
Electronic publication date: 2019 Sep 12
Publication date: 2019
Volume: 7
Electronic Location ID: e7664
Received 2019 Mar 2; Accepted 2019 Aug 13
Copyright: ©2019 Qian et al.
Copyright year: 2019
Copyright holder: Qian et al.
License: This is an open access article distributed under the terms of the Creative Commons Attribution License, which permits unrestricted use, distribution, reproduction and adaptation in any medium and for any purpose provided that it is properly attributed. For attribution, the original author(s), title, publication source (PeerJ) and either DOI or URL of the article must be cited.
License URL: https://creativecommons.org/licenses/by/4.0/

Keywords: Eutrema salsugineum, Aquaporin, Abiotic stress, Gene structure, Expression pattern

Funding: National Natural Science Foundation of China 31570396 Fundamental Research Funds for the Central Universities 2572016DA05 This work was supported by the National Natural Science Foundation of China (No. 31570396) and the Fundamental Research Funds for the Central Universities (2572016DA05). The funders had no role in study design, data collection and analysis, decision to publish, or preparation of the manuscript.

==============================
Aquaporins (AQPs) serve as water channel proteins and belong to major intrinsic proteins (MIPs) family, functioning in rapidly and selectively transporting water and other small solutes across biological membranes. Importantly, AQPs have been shown to play a critical role in abiotic stress response pathways of plants. As a species closely related to Arabidopsis thaliana, Eutrema salsugineum has been proposed as a model for studying salt resistance in plants. Here we surveyed 35 full-length AQP genes in E. salsugineum, which could be grouped into four subfamilies including 12 plasma membrane intrinsic proteins (PIPs), 11 tonoplast intrinsic proteins (TIPs), nine NOD-like intrinsic proteins (NIPs), and three small basic intrinsic proteins (SIPs) by phylogenetic analysis. EsAQPs were comprised of 237–323 amino acids, with a theoretical molecular weight (MW) of 24.31–31.80 kDa and an isoelectric point (pI) value of 4.73–10.49. Functional prediction based on the NPA motif, aromatic/arginine (ar/R) selectivity filter, Froger’s position and specificity-determining position suggested quite differences in substrate specificities of EsAQPs. EsAQPs exhibited global expressions in all organs as shown by gene expression profiles and should be play important roles in response to salt, cold and drought stresses. This study provides comprehensive bioinformation on AQPs in E. salsugineum, which would be helpful for gene function analysis for further studies.

Introduction

Water is the most abundant molecule in living cells, forming the basic medium in which all biochemical reactions take place (Dev & Herbert, 2018). Aquaporins (AQPs) belong to the major intrinsic proteins (MIPs) superfamily, which could selectively transport water molecules across the cell membrane. In addition, AQPs can also transport many small molecules, such as glycerol, urea, carbon dioxide (CO2), silicon, boron, ammonia (NH3) and hydrogen peroxide (H2O2) (Biela et al., 1999; Gerbeau et al., 1999; Uehlein et al., 2003; Ma et al., 2006; Takano et al., 2006; Loqué et al., 2005; Dynowski et al., 2008). AQP was first discovered in animals and subsequently found in almost all living organisms (Gomes et al., 2009). Compare to animals, plants have more robust and diverse AQPs. For instance, there are 35 AQPs in Arabidopsis thaliana, 33 in Oryza sativa, 40 in Sorghum bicolor, 72 in Glycine max, 47 in Cicer arietinum and 45 in Manihot esculenta (Johanson et al., 2001; Sakurai et al., 2005; Kadam et al., 2017; Zhang et al., 2013; Deokar & Tar’an, 2016; Putpeerawit et al., 2017).

Plant AQPs can be divided into seven subfamilies based on the protein sequence similarity analysis. Plasma membrane intrinsic proteins (PIPs) are the largest subfamily of plant AQPs. The most of the PIPs are commonly localized in the plasma membrane and are further divided into two phylogenetic groups PIP1 and PIP2. Tonoplast intrinsic proteins (TIPs) subfamily is usually localized in the tonoplast, which contain five classes TIP1, TIP2, TIP3, TIP4 and TIP5. NOD26-like intrinsic proteins (NIPs) named from NIP protein (Nodulin-26, GmNOD26), were discovered in the plasma membrane of soybean cells (Fortin, Morrison & Verma, 1987). Small basic intrinsic proteins (SIPs) are typically localized in the endoplasmic reticulum. X intrinsic proteins (XIPs) are present in some dicots but absent in Brassicaceae and monocots (Maurel et al., 2015). GlpF-like intrinsic proteins (GIPs) are found in moss (Physcomitrella patens) and similar to bacterial glycerol channels (Danielson & Johanson, 2008; Gustavsson et al., 2005). Hybrid intrinsic proteins (HIPs) are found in fern (Selaginella moellendorffii) and moss (Physcomitrella patens, Anderberg, Kjellbom & Johanson, 2012; Gustavsson et al., 2005). Therefore, some classes (such as XIPs, HIPs, or GIPs) are considered to be lost during the evolution of certain plant lineages due to function redundancies (Maurel et al., 2015).

AQPs are highly conserved in molecular structure, consisting of six transmembrane α-helical domains (TM1-TM6) linked by five loops (A-E), with both the N and C terminal having a cytoplasmic orientation. There are two highly conserved NPA (Asn-Pro-Ala) motifs in two half helices (HB and HE) of loopB and loopE at the center of the pore that have substrate selectivity (Taji et al., 2002). The narrow aromatic/arginine (ar/R) selectivity filter is formed with four residues from TM helix 2 (H2), TM helix 5 (H5), and loop E (LE1 and LE2), which has been shown to provide a size barrier for solute permeability (Bansal & Sankararamakrishnan, 2007). Froger’s position consists of five residues (P1-P5) that could transport two different types of molecules, water and glycerol (Froger et al., 1998). Moreover, it has been predicted that AQPs have nine specificity-determining positions (SDPs) for non-aqua substrates, such as ammonia, boron, carbon dioxide, hydrogen peroxide, silicon and urea, for each unique group (Hove & Bhave, 2011).

Salt cress previously named as Thellungiella halophila or Thellungiella salsuginea recently was corrected to Eutrema salsugineum based on taxonomy and systematics, which is a relative close to A. thaliana (Koch & German, 2013). As a salt-sensitive plant, Arabidopsis has certain limits to study the mechanism of salt and drought resistance. In contrast, E. salsugineum, with a small genome, is quite tolerant to salt, drought and low temperature stresses, being considered to be a halophyte model plant for investigating the mechanism of plant resistance to stress (Zhu, 2001; Inan et al., 2004). The E. salsugineum AQPs like TsTIP1;2, TsMIP6 and TsPIP1;1 have been found to play an important role in plant response to abiotic stress (Wang et al., 2014; Sun et al., 2015; Li et al., 2018). The E. salsugineum genome was sequenced in 2012 and 2013 at the chromosome level and scaffold level, respectively (Wu et al., 2012; Yang et al., 2013), promoting the bioinformatics analysis of whole aquaporin family.

In this study, a genome-wide analysis of AQP genes was carried out in E. salsugineum, a total of 35 full-length AQP genes were identified. Based on the phylogenetic analysis, we found that the identified EsAQPs were quite similar to AtAQPs. The EsAQPs could be grouped into four subfamilies, including PIPs, TIPs, NIPs and SIPs. Protein sequences, chromosome distributions, gene structures and putative functions were analyzed for each of these members. The expression level of EsAQP genes in different organs and the abundance change of EsAQP genes in response to salt, drought and cold stresses were also investigated.

Materials & Methods

Identification and chromosomal location of EsAQPs

The whole genome of E. salsugineum was downloaded from NCBI (https://www.ncbi.nlm.nih.gov/genome/12266, Wu et al., 2012; Yang et al., 2013). To identify E. salsugineum AQP candidate genes, a Hidden Markov Model (HMM) analysis was used. HMM profile of MIP (PF00230) was downloaded from Pfam protein family database (http://pfam.sanger.ac.uk/) and used as the query (P < 0.05) to search for AQP proteins in the E. salsugineum genome. To avoid missing potential AQP members, the NCBI BLAST tool was used to search E. salaugineum AQPs and known Arabidopsis AQP protein sequences as a query, and the top five aligned sequences were considered as candidates. After removing all of the redundant sequences, the sequences of putative EsAQP genes were loaded on relative chromosomes of E. salsugineum using the SnapGene tool. The map of the chromosome position of each EsAQP genes was drawn by MapInspect 1.0.

Classification, phylogenetic analysis and structural features

Multiple sequence alignments of putative AQP proteins were performed by ClustalW, and a phylogenetic tree was constructed using neighbor joining with MEGA 6.0 (Tamura et al., 2013). The transmembrane regions were detected using TOPCONS (http://topcons.cbr.su.se/pred/) and TMHMM (http://www.cbs.dtu.dk/services/TMHMM/). Protein subcellular localization of E. salsugineum AQPs was predicted in Plant-mPLoc (http://www.csbio.sjtu.edu.cn/bioinf/plant-multi/) and WoLF PSORT (http://www.genscript.com/wolf-psort.html). Functional predictions, such as NPA motifs, ar/R filters (H2, H5, LE1 and LE2), Froger’s positions (P1-P5) and nine specificity-determining positions (SDP1-SDP9), were analyzed by the alignments with function known AQPs (Quigley et al., 2001; Park et al., 2010; Hove & Bhave, 2011). The gene structure for each EsAQP was illustrated with the Gene Structure Display Server 2.0 (http://gsds.cbi.pku.edu.cn/). The conserved motifs of EsAQP proteins were analyzed by MEME suite (http://meme-suite.org/).

Plant materials and stress treatments

E. salsugineum seeds (ecotype Shandong, China) were provided by Prof. Hui Zhang (Shandong Normal University, Jinan, China). The seeds were plated on 1/2 MS medium and treated at 4 °C in dark for 7 days, then cultured in plant growth chamber with illumination of 150 µmol/m2/s, photoperiod 16/8 h of light/darkness at 25 °C and 60% relative humidity. After one week, the seedlings were transferred into a mixed medium with soil and vermiculite (3:1). Vernalization treatment for bolting was conducted in 4-week old seedlings at 4 °C for 4 weeks, and then they were moved back to the growth chamber until they grew flowers. Samples of roots, stems, leaves, flowers and siliques were collected, immediately frozen in liquid nitrogen and stored at −80 °C for further analysis.

For abiotic stress assays, the 4-week old seedlings were exposed to 300 mM NaCl for 24 h as a salt stress condition, treated at 4 °C for 24 h as cold stress, and not irrigated until the soil moisture content was less than 20% for 7 days as drought stress. The aerial part of seedling was collected for further analysis.

RNA extraction, cDNA synthesis and qRT-PCR

The total RNA was extracted using TRIzol reagent (Takara) following the manufacturer’s protocol. The quality of the RNA was determined using an ultraviolet spectrophotometer (BioMate 3S; Thermo Fisher). After removing genomic DNA contamination with DNase I, cDNA was synthesized by using the PrimeScript™ RT Reagent Kit (Takara). Three biological replicates of cDNA samples were used for qRT-PCR analysis with three technical replicates.

Primers of EsPIP genes were designed using Primer 3.0 (http://bioinfo.ut.ee/primer3-0.4.0/) and the reference gene was taken from Wang et al. (2014). All of primers were listed in Table S1. The qRT-PCR analysis was conducted in Applied Biosystems 7500 Real-Time PCR System (ABI, USA) by using SYBR Premix Ex TaqTM II (Takara). Reaction system contained 10 µl SYBR Premix Ex Taq II, 2 µl 5-fold diluted cDNA, 0.8 µl of each primer (10 mM), and ddH2O to a final volume of 20 µl. The PCR program was set as follows: 95 °C for 30 s, followed by 40 cycles of 95 °C for 5 s and 60 °C for 34 s. Then, a melting curve was generated to analyze the specificity of each primer with a temperature shift from 60 to 95 °C. The fold changes of the EsAQP genes expression under abiotic stresses were calculated with the 2−ΔΔCt method, while the gene expressions level of EsAQP genes in each organ were calculated with the ΔCt method. The heat map of gene expression pattern was visualized using HemI software.

Subcellular localization of EsPIP1;2 and EsPIP2;1 proteins

The coding sequences of EsPIP1;2 and EsPIP2;1 were amplified using primers containing the XbaI/SalI restriction site (Table S2). The purified products were subcloned into a reconstructed pBI121 vector which was composed of the XbaI/SalI site, and GFP. pBI121-EsPIP1;2-GFP and pBI121-EsPIP2;1-GFP vectors were transformed into A. tumefaciens strain GV3101. Then transient transformation in onion epidermis according to the method of Xu et al. (2014) took place. Images of epidermal cells were taken by fluorescence microscope with a mirror unit (U-BW).

Xenopus oocyte expression and osmotic water permeability assay

The coding regions of EsPIP1;2 and EsPIP2;1 were subcloned into pCS107 vector using the restriction sites BamHI and EcoRI (see primers in Table S2). After linearization, the cRNAs were synthesized in vitro using the Sp6 mMessage mMachine kit (Ambion). Oocyte preparation, injection, and expression were performed as described by Hu et al. (2012) with a little modification. A total of 10 nl water or cRNAs of EsPIP1;2 and EsPIP2;1 (one ng/nl) were injected into oocytes, respectively, and then the oocytes were incubated at 18 °C for 48 h in Oocyte Culture Medium (OCM, 50% L-15, 40% HEPES (pH 7.4), 10% calf serum, 0.5% penicillin and 10 mg/ml streptomycin).The osmotic water permeability coefficient of oocytes was determined as described by Zhang & Verkman (1991). To measure the osmotic water permeability coefficient, oocytes were transferred to 5-fold diluted OCM solution. Changes in the oocytes volume were monitored at room temperature with a microscope video system. Oocytes volumes (Vs) were calculated from the measured area of each oocyte. The osmotic Pf was calculated for the Erst 10 min using the formula Pf = V 0[d(V∕V 0)∕dt]∕[S0 × V w(Osmin–Osmout)]. V 0 and S0 are the initial volume and surface area of each individual oocyte, respectively; d(V/V0)/dt is the relative volume increase per unit time; Vw is the molar volume of water (18 cm3 mol−1); and Osmin–Osmout is the osmotic gradient between the inside and outside of the oocyte.

Results

Characterization, classification and chromosome localization of EsAQPs

To extensively identify AQPs in E. salsguineum, HMM profile of the MIP domain (PF00230) was used. As a result, a total of 35 putative EsAQPs were identified for further analysis (Table 1). To classify the AQP members, a phylogenetic tree was constructed according to the similarity of AQP protein sequences in E. salsugineum and Arabidopsis through the neighbor-joining method (Fig. 1). Based on the phylogenetic analysis, we found that the identified EsAQPs have very high similarity with AtAQPs which can be grouped into four subfamilies, including 12 PIPs, 11 TIPs, nine NIPs and three SIPs. In addition, the EsPIP subfamily was further divided into two classes (five EsPIP1s and seven EsPIP2s), the EsTIP subfamily into five classes (three EsTIP1s, four EsTIP2s, two EsTIP3s, one EsTIP4s and one EsTIP5s), the EsNIP subfamily into seven classes (1 EsNIP1s, 1 EsNIP2s, 1 EsNIP3s, three EsNIP4s, one EsNIP5s, one EsNIP6s and one EsNIP7s), and the EsSIP subfamily into two classes (two EsSIP1s and one EsSIP2s). The nomenclature of E. salsugineum AQPs was based on their phylogenetic relationship with AtAQPs (Fig. 1). These results were also supported by the existing annotation for E. salsugineum obtained from Phytozyme (https://phytozome.jgi.doe.gov/pz/portal.html#!bulk?org=Org_Esalsugineum), most of our identified EsAQPs were matched to the existing annotations (Table S3). In addition, we have updated some annotations i.e., Thhalv10008397m and Thhalv10025910m, which were both annotated as PIP1;4 in Phytozyme, were renamed with EsPIP1;3 and EsPIP1;5, respectively, and other details were listed in Table S3. Compare to AtAQPs, PIP2;8 and NIP1;1 were missing in E. salsugineum. And TIP2;4 and NIP4;3 were identified in E.salsugineum, but not found in Arabidopsis, which were shared high similarity with their homologous genes. In Arabidopsis, 35 AQP genes were unevenly distributed on the five chromosomes Feng et al. (2017). As shown in Table 1 and Fig. 2, the chromosomal locations of 34 EsAQP genes were randomly assigned to all the seven chromosomes. However, chromosomal location of EsTIP2;2 could not be determined. Overall, AQPs from E. salsugineum had a very close relationship with those from Arabidopsis.

Table 1 Details of EsAQP genes identified from the genome-wide search analysis.

Name	Chromosomal Localization	Scaffold	Coding sequence	Protein ID	Plant-mPLoc	WoLF PSORT	Plant species	Subcellular localization	Reference	
EsPIP1;1	Chr5;748,014∼746,287	NW_006256838.1	XM_006402419.1	XP_006402482.1	plas	plas	Oryza ativa	plas	Liu et al. (2013)	
EsPIP1;2	Chr4;24,198,933∼24,200,732	NW_006256812.1	XM_006397718.1	XP_006397781.1	plas	plas	Musa nana	plas	Sreedharan, Shekhawat & Ganapathi (2013)	
EsPIP1;3	Chr1;227,418∼229,068	NW_006256612.1	XM_006418376.1	XP_006418439.1	plas	plas				
EsPIP1;4	Chr6;182,520∼180,408	NW_006256756.1	XM_006396178.1	XP_006396241.1	plas	plas	Arabidopsis thaliana	plas	Li et al. (2015)	
EsPIP1;5	Chr7;21,955,256∼21,956,964	NW_006256909.1	XM_006413496.1	XP_006413559.1	plas	plas				
EsPIP2;1	Chr5;3,815,044∼3,817,131	NW_006256858.1	XM_006403628.1	XP_006403691.1	plas	plas	A. thaliana	plas	Li et al. (2011)	
EsPIP2;2	Chr4;20,408,518∼20,407,373	NW_006256908.1	XM_006410833.1	XP_006410896.1	plas	plas	Vitis vinifera	plas	Leitäo et al. (2012)	
EsPIP2;3	Chr4;20,411,864∼20,413,318	NW_006256908.1	XM_006410834.1	XP_006410897.1	plas	plas				
EsPIP2;4	Chr6;21,418,342∼21,416,629	NW_006256829.1	XM_006400761.1	XP_006400824.1	plas	plas	Zea mays	plas	Zelazny et al. (2009)	
EsPIP2;5	Chr5;3,318,416∼3,315,956	NW_006256858.1	XM_006403468.1	XP_006403531.1	plas	plas	Z. mays	plas	Zelazny et al. (2009)	
EsPIP2;6	Chr4;21,319,556∼21,322,584	NW_006256908.1	XM_006411061.1	XP_006411124.1	plas	plas	M. nana	plas	Sreedharan, Shekhawat & Ganapathi (2015)	
EsPIP2;7	Chr7;27,180,960∼27,182,785	NW_006256909.1	XM_006412089.1	XP_006412152.1	plas	plas	A. thaliana	plas	Hachez et al. (2014)	
EsTIP1;1	Chr4;20,182,942∼20,184,210	NW_006256908.1	XM_006410791.1	XP_006410854.1	vacu	cyto	A. thaliana	vacu	Ma et al. (2004)	
EsTIP1;2	Chr2;16,508,526∼16,506,789	NW_006256547.1	XM_006395487.1	XP_006395549.1	vacu	plas/vacu	Eutrema salsiguneum	vacu	Wang et al. (2014)	
EsTIP1;3	Chr6;663,103∼662,130	NW_006256756.1	XM_006396285.1	XP_006396348.1	vacu	cyto				
EsTIP2;1	Chr3;5,624,419∼5,626,413	NW_006256885.1	XM_006406794.1	XP_006406857.1	vacu	chlo/vacu	A. thaliana	vacu	Loque et al. (2005)	
EsTIP2;2	NA	NW_006256909.1	XM_006414179.1	XP_006414242.1	vacu	vacu	Triticum aestivum	vacu	Chunhui et al. (2013)	
EsTIP2;3	Chr2;14,894,399∼14,893,306	NW_006256828.1	XM_006398375.1	XP_006398438.1	vacu	vacu	A. thaliana	vacu	Loque et al. (2005)	
EsTIP2;4	Chr1;27,709,976∼27,708,236	NW_006256486.1	XM_006392888.1	XP_006392950.1	vacu	vacu				
EsTIP3;1	Chr5;22,490,388∼22,491,488	NW_006256342.1	XM_006390520.1	XP_006390582.1	vacu	chlo/cyto/vacu	A. thaliana	plas/vacu	Gattolin, Sorieul & Frigerio (2011)	
EsTIP3;2	Chr1;6,309,744∼6,311,048	NW_006256612.1	XM_006416602.1	XP_006416665.1	vacu	chlo/mito/vacu	A. thaliana	plas/vacu	Gattolin, Sorieul & Frigerio (2011)	
EsTIP4;1	Chr4;7,484,947∼7,486,691	NW_006256895.1	XM_006408738.1	XP_006408801.1	vacu	vacu				
EsTIP5;1	Chr5;6,934,814∼6,933,858	NW_006256858.1	XM_006404316.1	XP_006404379.1	vacu / plas	chlo	A. thaliana	mito	Soto et al. (2010)	
EsNIP1;2	Chr7;19,890,089∼19,892,520	NW_006256909.1	XM_006413978.1	XP_006414041.1	plas	plas	A. thaliana	plas	Wang et al. (2017)	
EsNIP2;1	Chr4;19,043,681∼19,042,522	NW_006256908.1	XM_006410521.1	XP_006410584.1	plas	vacu:	A. thaliana	plas/E.R	Choi & Roberts (2007), Mizutani et al. (2006)	
EsNIP3;1	Chr1;12,292,410∼12,294,335	NW_006256612.1	XM_006415218.1	XP_006415281.1	plas	vacu	O. sativa	plas	Hanaoka et al. (2014)	
EsNIP4;1	Chr7;4,484,562∼4,482,986	NW_006256877.1	XM_006405767.1	XP_006405830.1	plas	plas	A. thaliana	plas/vacu	DiGiorgio et al. (2016)	
EsNIP4;2	Chr7;4,513,301∼4,511,485	NW_006256877.1	XM_006405768.1	XP_006405831.1	plas	plas	A. thaliana	plas/vacu	DiGiorgio et al. (2016)	
EsNIP4;3	Chr7;4,481,446∼4,479,745	NW_006256877.1	XM_006405766.1	XP_006405829.1	plas	plas				
EsNIP5;1	Chr6;6,005,178∼6,008,910	NW_006256756.1	XM_006397006.1	XP_006397069.1	plas	plas	A. thaliana	plas	Takano et al. (2006)	
EsNIP6;1	Chr5;25,383,958∼25,386,014	NW_006256342.1	XM_006389768.1	XP_006389830.1	plas	plas	A. thaliana	plas	Tanaka et al. (2008)	
EsNIP7;1	Chr3;1,929,290∼1,927,201	NW_006256885.1	XM_006407920.1	XP_006407983.1	plas	cyto				
EsSIP1;1	Chr3;1,105,251∼1,102,416	NW_006256885.1	XM_024159977.1	XP_024015745.1	plas	plas	A. thaliana	E.R	Ishikawa et al. (2005)	
EsSIP1;2	Chr6;23,161,081∼23,162,581	NW_006256829.1	XM_006400314.1	XP_006400377.1	vacu plas	vacu	A. thaliana	E.R	Ishikawa et al. (2005)	
EsSIP2;1	Chr5;2,401,441∼2,403,463	NW_006256838.1	XM_006402867.1	XP_006402930.1	plas	E.R	A. thaliana	E.R	Ishikawa et al. (2005)	
Notes.

Abbreviationplas plasma membrane

cyto cytosol

vacu tonoplast membrane

chlo chloroplast

mito mitochondria

E.R endoplasmic reticulum

NA not applicable

Figure 1 Phylogenetic tree of AQP amino acid sequences from E. salsugineum and A. thaliana.

Alignments were performed using the default parameter of ClustalW and the phylogenetic tree was constructed using the Neighbor-Joining tree method with 1,000 bootstrap replicates in MEGA6.0 software. Each subfamily of AQPs was well separated in different clades and represented by different colors. The solid circle represents EsAQPs and the hollow circle represents AtAQPs.

Figure 2 Chromosomal localization of the EsAQP genes.

The diagram was drawn using the MapInspect software, and 34 out of 35 EsAQPs were located on seven chromosomes (except EsTIP2;2).

Gene structure and subcellular localization analysis of EsAQPs

Gene structure analysis of the 35 EsAQP genes was performed on the Gene Structure Display Server of NCBI. Based on their mRNA and genomic DNA sequences, we found exon lengths were mostly conserved in each subfamily of EsAQP gene with same exon number, but introns varied in both length and position (Fig. 3). All members of the EsPIP subfamily contained four exons with similar length (289–328, 296, 141 and 93–126 bp, respectively) and conserved sequences in the 2nd and 3rd exon, except for EsPIP2;4, which had a shorter 2nd and longer 3rd exon (307, 151, 286, and 111 bp). The majority members of the EsTIP subfamily contained three exons with similar lengths, and the other members had two exons with similar lengths, except for EsTIP1;3, which had only one exon without intron. In the EsNIP subfamily, some members exhibited five exons with similar lengths, while others had four exons with varied lengths. All EsSIP subfamily genes displayed three exons with similar lengths. This description of exon-intron structure provides additional evidence to support the classification results (Kong et al., 2017).

Figure 3 Gene structures of the EsAQP genes.

The blue rectangle, yellow rectangle and black line represent UTR, exon and intron, respectively.

The prediction of subcellular localization showed diverse results, not always in agreement with experimentally determined localizations (reviewed in Katsuhara et al., 2008). In summary, the prediction of EsAQP subcellular localization in Plant-mPLoc showed that EsPIP, EsNIP and EsSIP subfamilies were localized in plasma membrane, while EsTIP subfamily members were localized in tonoplast membrane. Among them, EsPIP1;2 and EsTIP5;1 were localized in both tonoplast membrane and plasma membrane (Table 1). Moreover, WoLF PSORT predicts different location for EsAQPs and assigns values for that location (Table S4). The highest values list in Table 1 showed that EsPIPs were predicted to localize in plasma membrane, which were consistent with the Plant-mPLoc prediction and many other reports (Cui et al., 2008; Hu et al., 2012; Xu et al., 2014). The majority of other AQP members were predicted to localize in plasma membrane or tonoplast membrane, except for EsTIP5;1, EsNIP7;1 and EsSIP2;1, which were predicted to be associated with chloroplast, cytosol and endoplasmic reticulum, respectively. Moreover, some members showed multiple type of localization; for example, EsTIP3;1 was predicted to be associated with chloroplast/cytosol/tonoplast membrane and EsTIP3;2 with chloroplast/mitochondria/tonoplast membrane. The subcellular localization of most published AQP homologous was consistent with the predicted results in E. salsugineum (Table 1). These observations demonstrated that the subcellular localization of AQPs may be complex and diverse.

To verify the predictions, genes of EsPIP1;2 and EsPIP2;1 were cloned into the pBI121-GFP vector to create the 35S::EsPIP-GFP fusion proteins. The plasmid was transformed into onion epidermis by agrobacterium-mediated transformation. As shown in Fig. 4, the GFP fluorescence mainly exhibit in plasma membrane, indicated that EsPIP1;2 and EsPIP2;1 proteins were consistent with the predictions. Although not conclusive, the predicted localization could serve as a useful reference for further studies on EsAQPs protein functions in plants.

Figure 4 Subcelluar localizations of EsPIP1;2 and EsPIP2;1 proteins.

Onion epidermal cells transiently transformed with empty vector (A, B), EsPIP1;2-GFP (C, D) and EsPIP2;1-GFP (E, F), respectively. The images were visualized under fluorescence microscope. A, C, E: bright-field images; B, D, F: green fluorescence images.

Structure characteristics of EsAQPs

Sequence analysis showed that all EsAQPs contain six transmembrane domains (TMDs) comprising 237–323 amino acids, had theoretical molecular weights (MW) of 24.31–31.80 kDa and isoelectric point (pI) values of 4.73–10.49 (Table 2). The EsPIP subfamily had a similar molecular weight of approximately 30.84 kDa. Most members of the EsNIP subfamily exhibited a similar molecular weight and isoelectric point of EsPIP subfamily. The EsTIP and EsSIP subfamilies had lower MW among the EsAQPs, and the isoelectric points of these two subfamilies were acidic and alkaline, respectively (Fig. S1).

Table 2 Structural characteristics of the EsAQPs.

Name	AA	TM	MW(KD)	pI	NPA motif	ar/R selectivity filter	Froger’s positions	
					LB	LE	H2	H5	LE1	LE2	P1	P2	P3	P4	P5	
PIPs															
EsPIP1;1	286	6	30.77	9.14	NPA	NPA	F	H	T	R	Q	S	A	F	W	
EsPIP1;2	286	6	30.60	9.16	NPA	NPA	F	H	T	R	Q	S	A	F	W	
EsPIP1;3	286	6	30.62	9.02	NPA	NPA	F	H	T	R	Q	S	A	F	W	
EsPIP1;4	286	6	30.56	9.02	NPA	NPA	F	H	T	R	Q	S	A	F	W	
EsPIP1;5	287	6	30.61	9.00	NPA	NPA	F	H	T	R	Q	S	A	F	W	
EsPIP2;1	287	6	30.48	6.95	NPA	NPA	F	H	T	R	Q	S	A	F	W	
EsPIP2;2	284	6	30.21	6.50	NPA	NPA	F	H	T	R	Q	S	A	F	W	
EsPIP2;3	285	6	30.31	6.51	NPA	NPA	F	H	T	R	Q	S	A	F	W	
EsPIP2;4	285	6	30.12	7.62	NPA	NPA	F	H	T	R	Q	S	A	F	W	
EsPIP2;5	286	6	30.57	8.82	NPA	NPA	F	H	T	R	Q	S	A	F	W	
EsPIP2;6	290	6	31.11	7.69	NPA	NPA	F	H	T	R	Q	S	A	F	W	
EsPIP2;7	281	6	29.82	9.11	NPA	NPA	F	H	T	R	M	S	A	F	W	
TIPs																
EsTIP1;1	251	6	25.62	6.03	NPA	NPA	H	I	A	V	T	A	A	Y	W	
EsTIP1;2	253	6	25.70	5.32	NPA	NPA	H	I	A	V	T	A	A	Y	W	
EsTIP1;3	252	6	25.85	5.10	NPA	NPA	H	I	A	V	T	S	A	Y	W	
EsTIP2;1	277	6	28.32	7.80	NPA	NPA	H	I	G	R	T	S	A	Y	W	
EsTIP2;2	250	6	25.02	4.87	NPA	NPA	H	I	G	R	T	S	A	Y	W	
EsTIP2;3	243	6	24.31	4.73	NPA	NPA	H	I	G	R	T	S	A	Y	W	
EsTIP2;4	254	6	25.85	5.43	NPA	NPA	H	I	G	R	T	S	A	Y	W	
EsTIP3;1	265	6	27.94	7.17	NPA	NPA	H	T	A	R	T	A	A	Y	W	
EsTIP3;2	267	6	28.29	6.58	NPA	NPA	H	M	A	R	T	T	A	Y	W	
EsTIP4;1	249	6	26.16	5.49	NPA	NPA	H	I	A	R	T	S	A	Y	W	
EsTIP5;1	257	6	26.70	7.72	NPA	NPA	N	V	G	C	V	A	A	Y	W	
NIPs																
EsNIP1;2	297	6	31.80	8.83	NPA	NPA	W	V	A	R	F	S	A	Y	L	
EsNIP2;1	286	6	30.56	6.78	NPA	NPG	W	V	A	R	F	S	A	Y	L	
EsNIP3;1	323	6	34.46	5.94	NPA	NPA	W	I	A	R	F	S	A	Y	L	
EsNIP4;1	283	6	30.49	8.73	NPA	NPA	W	V	A	R	F	S	A	Y	L	
EsNIP4;2	284	6	30.34	8.80	NPA	NPA	W	V	A	R	F	S	A	Y	L	
EsNIP4;3	283	6	30.30	8.98	NPA	NPA	W	V	A	R	F	S	A	Y	L	
EsNIP5;1	301	6	31.20	8.31	NPS	NPA	A	I	G	R	F	T	A	Y	L	
EsNIP6;1	305	6	31.78	8.57	NPA	NPA	A	I	A	R	F	T	A	Y	L	
EsNIP7;1	275	6	28.62	6.12	NPS	NPA	A	V	G	R	Y	S	A	Y	L	
SIPs																
EsSIP1;1	238	6	25.41	9.89	NPT	NPA	I	V	P	I	I	A	A	Y	W	
EsSIP1;2	242	6	25.96	9.83	NPC	NPA	V	F	P	I	I	A	A	Y	W	
EsSIP2;1	237	6	25.85	9.64	NPL	NPA	S	H	G	A	F	V	A	Y	W	
Notes.

AbbreviationAA amino acids length

TM transmembrane domain

MW molecular weight

pI isoelectricpoint NPA Asn-Pro-Ala motif

ar/R aromatic/arginine

NPA motifs, ar/R selectivity filters and Froger’s positions of AQP protein sequences play critical role in channel selectivity. The sequence alignment between AtAQPs and GhAQPs was carried out to analyze the conserved domains (Quigley et al., 2001; Park et al., 2010). The results in Table 2 showed that all EsPIP subfamily members had two typical NPA motifs in loop B and loop E, with a water transport ar/R filter with amino acid of F-H-T-R. Froger’s position consists of Q-S-A-F-W in most cases, except for EsPIP2;7, which had an M at the P1 position. All EsTIP subfamily had two typical NPA motifs. The ar/R was composed of H–I-A-V in EsTIP1s, H-I-G-R in EsTIP2s and H-T/M/I-A-R in other EsTIP members, while in EsTIP5;1, it was composed of N-V-G-C. Froger’s position consists of T-A/S-A-Y-W, except for EsTIP5;1 and EsTIP3;2, which had a V at the P1 position and a T at the P2 position respectively. Most members of EsNIP subfamily had two typical NPA motifs, not in EsNIP2;1 (with an NPG in LE), EsNIP5;1 and EsNIP7;1 (with an NPS in LB). The ar/R filter consists of residues like W/A-V/I-A/G-R, and Froger’s position consists of F-S-A-Y-L, except for EsNIP7;1, which had a Y at the P1 position, and for EsNIP5;1 and EsNIP6;1 had a T at the P2 position. The EsSIP subfamily showed a variable site in the first NPA, the alanine (A) was replaced by threonine (T), cysteine (C) or leucine (L). The ar/R filter was also inconsistent with each other: I-V-P-I in EsSIP1;1, V-F-P-I in EsSIP1;2 and S-H-G-A in EsSIP2;1. The Forger’s position was composed of I-A-A-Y-W in EsSIP1s, while it was F-V-A-Y-W in EsSIP2;1.

MEME (Multiple EM for Motif Elicitation) is one of the most widely used tools for searching for novel “signals” in sets of biological sequences, include the discovery of new transcription factor binding sites and protein domains (Bailey et al., 2006). Conserved motifs of EsAQP proteins were predicted by MEME suite (Fig. 5). The result showed that motif 1, 2, 3, 4, 7, 8, and 10 were same in all EsPIPs, and motif 2, 4, 7, and 10 were unique. In addition, motif 9 was unique in EsPIP1s and can be used to distinguish EsPIP1s from EsPIP2s. This pattern of conserved motifs in the PIP subfamily also occurs in other plants and PIP1s contain one unique motif (Tao et al., 2014; Yuan et al., 2017). In the EsTIP subfamily, almost all EsTIPs had two motif 1, two motif 3, one motif 5 and one motif 6. Except for EsTIP1;3, which had no motif 6. Motif 5 could be an identifier of EsTIPs among the AQPs of E. salsugineum except for EsTIP5;1. Most of members in NIP subfamily had two motif 1, two motif 3, and two motif 6, except for EsNIP2;1 (lose one motif 1), EsNIP3;1 (lose one motif 6) and EsNIP5;1 (lose one motif 3). The two motif 6 might be used to distinguish EsNIPs with other EsAQPs. All EsSIP subfamily carried motif 3. Motif 8 appeared in EsSIP1s but not in EsSIP2;1, so it might be an specific trait of this group. This is a common phenomenon in plant SIP subfamily contains less motifs (Tao et al., 2014; Reddy et al., 2015; Yuan et al., 2017; Kong et al., 2017). Based on these analysis, it was evident that there were structural differences in various EsAQP subfamilies, but conserved in their own subfamily.

Figure 5 Conversed motif analysis in EsAQPs.

The conversed motif prediction was identified using MEME motif search analysis, and the maximum number parameter was set to 10. Different motifs were represented by different colors. (A) Conversed motifs of 35 EsAQP proteins correspond to p-values. (B) Motif consensus sequences.

Expression pattern of EsAQPs

The expression of EsAQP genes in different organs, including root, stem, leaf, flower and silique, was analyzed by RT-qPCR. The results showed that 35 EsAQP genes were detected in all the organs (Fig. 6A). Almost all EsPIP genes were highly expressed in all organs, except for EsPIP2;5 in leaf. In addition, the EsPIP genes, EsTIP1;1, EsTIP1;2, EsNIP1;2, EsNIP5;1, EsSIP1;1 and EsSIP2;1 were also highly expressed in all organs. Some EsAQP genes, such as EsTIP2;3, EsTIP2;4, EsNIP2;1 and EsNIP3;1, were specifically highly expressed in root. Two EsTIPs (EsTIP2;2 and EsTIP5;1), three EsNIPs (EsNIP4;1, EsNIP4;3 and EsNIP7;1) and EsSIP1;2 were highly expressed only in flower. Two EsTIPs (EsTIP3;1 and EsTIP3;2) were expressed in silique with relative high abundance. Compared analysis of each EsAQP gene between different organs revealed that most EsAQP genes showed higher expression level in flower than in other organs.

Figure 6 Expression profiles of the EsAQP genes.

(A) EsAQP genes expression in response to abiotic stress. The color scale represents the 2−ΔΔCt value normalized to untreated controls and log2 transformed counts, where green indicates downregulated expression and red indicates upregulated expression. (B) Expression of EsAQP genes in various organs of E. salsugineum. Color scales represent 2ΔCt values normalized to actin and log2 transformed counts, where green indicates low expression and red indicates high expression.

Abiotic stresses are the main limiting factors for plants during environmental conditions that induce osmotic stress and disturb water balance. AQPs play major roles in maintaining water homeostasis and responding to environmental stresses in plants. Therefore, we further investigated the expression patterns of EsAQP genes under salt, drought and cold stress by qRT-PCR. The results showed that most of the EsAQP genes were up-regulated under salt and cold stress but down-regulated under drought stress (Fig. 6B). We found that five EsAQP genes were up-regulated under all the types of abiotic stresses, including EsPIP2;4, EsTIP1;2, EsNIP4;3, EsNIP5;1 and EsSIP1;2, while three EsAQP genes were down-regulated under all the types of abiotic stresses, including EsPIP1;5, EsTIP2;2 and EsTIP2;4. In addition, EsPIP1;1 and EsPIP2;2 were specifically up-regulated under salt stress, and EsPIP2;1, EsTIP2;1, EsTIP5;1, EsNIP4;1 and EsNIP6;1 were up-regulated only under cold stress.

Water permeability of EsPIP1;2 and EsPIP2;1

Previously, AtPIP2;1 has been reported that is an integral membrane protein that facilitates water transport across plasma membrane while AtPIP1;2 has no function (Li et al., 2011; Heckwolf et al., 2011). To determine the water channel activity of EsPIP1;2 and EsPIP2;1, proteins were tested in the Xenopus oocyte system. After two days of cRNA or water injection, the change rate in oocyte volume (Fig. 7A) and the osmotic water permeability coefficient (Pf) (Fig. 7B) were calculated. Expression of EsPIP2;1 conferred a rapid osmotically driven increase in relative volume, while expression of EsPIP1;2 enabled an increase in relative volume at a slower rate than the water-injected oocytes. Compared with water-injected control, the oocytes expressing EsPIP1;2 and EsPIP2;1 showed 1.39-fold and 2.08-fold increase in Pf, suggesting that both EsPIP1;2 and EsPIP2;1 are functional AQP with water channel activity. Meanwhile, our result is consistent with the known information that PIP2s have high efficiency water transfer activity but PIP1s have little or no increase in the Pf (Chaumont & Tyerman, 2014).

Figure 7 Water channel activity appraisals of EsPIP1;2 and EsPIP2;1.

(A) The swelling rates of Xenopus oocytes injected with H2O, or cRNA encoding EsPIP1;2 and EsPIP2;1, respectively. The rate of oocyte swelling upon immersion in hypo-osmotic medium is drawn as V/V0, where V is the volume at a given time point and V0 is the initial volume. (B) Water permeability codfficient (Pf) of oocytes injected with cRNA encoding H2O, or EsPIP1;2, or EsPIP2;1. The Pf values were calculated from the rate of oocyte swelling. Vertical bars indicate the SE. Asterisks indicate significant differences in comparison with oocytes injected with water. Statistical analysis were performed by SPSS 16.0 using one-way ANOVA and Least Significant Difference (LSD) test to detect significant differences (*p < 0.05, **p < 0.01).

Discussion

Gene duplication is a ubiquitous event that plays an important role in biological evolution, which may also contribute to stress tolerance via gene dosage increasing, avoiding some deleterious mutations and creating the opportunity for new function emergence (Innan & Kondrashov, 2010). AQPs are abundant, diverse and widely distributed in plants and involved in regulation of plant growth and development. From algae (two in Thalassiosira pseudonana and five in Phaeodactylum tricornutum) (Armbrust et al., 2004; Bowler et al., 2008) to fern (19 in Selaginella moellendorffii) (Danielson & Johanson, 2008) and moss (23 in Physcomitrella patens) (Anderberg, Kjellbom & Johanson, 2012) to higher plants (35 AQPs in Arabidopsis, 33 in Oryza sativa, 72 in Glycine max) (Johanson et al., 2001; Sakurai et al., 2005; Zhang et al., 2013), the number of AQPs has largely increased with evolution. Here, we provide a genome-wide information of AQP family of E. salsugineum.

In previous studies, it was shown that more than 95% gene families are shared in T. salsuginea (the former name of E. salsugineum, Koch & German, 2013) with A. thaliana (Wu et al., 2012) or more than 80% E. salsugineum genes had high homology orthologs in A. thaliana (Yang et al., 2013). The number of AQPs identified in E. salsugineum is the same as that in A. thaliana, and their protein sequences have very high similarities. No homologies of AtAQPs, PIP2;8 and NIP1;1 were not identified in E. salsugineum, while another two AQPs, TIP2;4 and NIP4;3 were found instead, which were not existed in Arabidopsis. These differences may not be directly illustrated the superiority of E. salsugineum in stress resistance, the functions of EsAQPs in resistance need to be further studied.

Structural analysis and functional inference of EsAQPs

Exon-intron structural divergence commonly happened in duplicate gene evolution and even in sibling paralogs; these changes occurred through the mechanisms of gain/loss, exonization/pseudoexonization and insertion/deletion (Xu et al., 2012). In common bean (Phaseolus vulgaris L.), each aquaporin subfamily are completely conserved in number, order and length of exons but varies in introns (Ariani & Gepts, 2015). The MEME motifs of the AQPs were conserved in all subfamilies, while a few were deleted, unique or family-specific, and a previous report also found this pattern in ZmPIPs (Bari et al., 2018). In our study, the exon-intron structure of EsAQP genes and the conserved MEME motifs of EsAQP protein sequences showed some common patterns (Figs. 3 and 5). These results indicated that the gene structure and the conserved motifs of EsAQPs shown subfamily-specific, these traits may provide new evidence to support the classification.

Table 3 Identified typical SDPs in EsAQPs.

The red font represent novel site.

Aquaporin	Specificity-determining positions	
	SDP1	SDP2	SDP3	SDP4	SDP5	SDP6	SDP7	SDP8	SDP9	
Ammonia Transporters	F/T	K/L/N/V	F/T	V/L/T	A	D/S	A/H/L	E/P/S	A/R/T	
EsTIP2;1	T	L	T	V	A	S	H	P	A	
EsTIP3;1	T	L	G	T	A	S	H	P	A	
EsNIP1;2	F	K	F	T	G	D	L	E	T	
EsNIP4;1	F	T	F	T	A	D	L	E	T	
EsNIP4;3	F	T	F	T	A	D	L	E	T	
Boric Acid transporter	T/V	I/V	H/I	P	E	I/L	I/L/T	A/T	A/G/P/K	
EsPIP1;1	T	I	H	P	E	L	L	T	P	
EsPIP1;2	T	I	H	P	E	L	L	T	P	
EsPIP1;3	T	I	H	P	E	L	L	T	P	
EsPIP1;4	T	I	H	P	E	L	L	T	P	
EsPIP1;5	T	I	H	P	E	L	L	T	P	
EsPIP2;5	T	I	H	P	E	L	L	T	P	
EsNIP5;1	T	I	H	P	E	L	L	A	P	
EsNIP6;1	T	I	H	P	E	L	L	A	P	
EsNIP7;1	V	I	H	P	E	L	L	T	P	
CO2transporter	I/L/V	I	C	A	I/V	D	W	D	W	
EsPIP1;1	L	I	C	A	I	D	W	D	W	
EsPIP1;2	V	I	C	A	I	D	W	D	W	
EsPIP1;3	V	M	C	A	I	D	W	D	W	
EsPIP1;4	V	M	C	A	I	D	W	D	W	
EsPIP1;5	V	I	C	A	I	D	W	D	W	
EsPIP2;4	V	I	C	A	V	E	W	D	W	
H2O2transporters	A/S	A/G	L/V	A/F/L/V/T	I/L/V	H/I/L/Q	F/Y	A/V	P	
EsPIP1;1	A	G	V	F	I	H	F	V	P	
EsPIP1;2	A	G	V	F	I	H	F	V	P	
EsPIP1;3	A	G	V	F	I	H	F	V	P	
EsPIP1;4	A	G	V	F	I	H	F	V	P	
EsPIP1;5	A	G	V	F	I	H	F	V	P	
EsPIP2;1	A	G	V	F	I	H	F	V	P	
EsPIP2;2	A	G	V	F	I	H	F	V	P	
EsPIP2;3	A	G	V	F	I	H	F	V	P	
EsPIP2;4	A	G	V	F	I	Q	F	V	P	
EsPIP2;5	A	G	V	F	I	H	F	V	P	
EsPIP2;6	A	G	V	F	I	Q	F	V	P	
EsPIP2;7	A	G	V	F	I	H	F	V	P	
EsTIP1;1	S	A	L	A	I	H	Y	A	P	
EsTIP1;2	S	A	L	A	I	H	Y	A	P	
EsTIP1;3	A	A	L	S	I	H	Y	V	P	
EsTIP2;1	S	A	L	V	I	H	Y	V	P	
EsTIP2;2	S	A	L	V	I	I	Y	V	P	
EsTIP2;3	S	A	L	V	I	I	Y	V	P	
EsTIP3;2	A	A	L	A	I	H	Y	V	P	
EsTIP4;1	S	A	L	L	T	H	Y	V	P	
EsNIP1;2	S	A	L	L	V	I	Y	V	P	
EsNIP3;1	S	A	L	V	I	L	Y	V	P	
EsNIP5;1	S	A	L	V	V	L	Y	V	P	
Silicic acid transporters	C/S	F/Y	A/E/L	H/R/Y	G	K/N/T	R	E/S/T	A/K/P/T	
Not found										
Urea Transporters	H	P	F/I/L/T	A/C/F/L	L/M	A/G/P	G/S	G/S	N	
EsPIP1;1	H	P	F	F	L	P	G	G	N	
EsPIP1;2	H	P	F	F	L	P	G	G	N	
EsPIP1;3	H	P	F	F	L	P	G	G	N	
EsPIP1;4	H	P	F	F	L	P	G	G	N	
EsPIP1;5	H	P	F	F	L	P	G	G	N	
EsPIP2;1	H	P	F	F	L	P	G	G	N	
EsPIP2;2	H	P	F	F	L	P	G	G	N	
EsPIP2;3	H	P	F	F	L	P	G	G	N	
EsPIP2;4	H	P	F	F	L	P	G	G	N	
EsPIP2;5	H	P	F	F	L	P	G	G	N	
EsPIP2;6	H	P	F	F	L	P	G	G	N	
EsPIP2;7	H	P	F	F	L	P	G	G	N	
EsTIP1;1	H	P	F	F	L	A	G	S	N	
EsTIP1;2	H	P	F	F	L	A	G	S	N	
EsTIP1;3	H	P	F	F	L	A	G	S	N	
EsTIP2;1	H	P	F	A	L	P	G	S	N	
EsTIP2;2	H	P	L	A	L	P	G	S	N	
EsTIP2;3	H	P	L	A	L	P	G	S	N	
EsTIP2;4	H	P	F	V	L	P	G	S	N	
EsTIP3;1	H	P	F	L	L	P	G	S	N	
EsTIP3;2	H	P	L	L	L	P	G	S	N	
EsTIP4;1	H	P	I	L	L	A	G	S	N	
EsTIP5;1	H	P	F	A	L	P	G	S	N	
EsNIP1;2	H	P	I	A	L	P	G	S	N	
EsNIP2;1	H	P	I	A	L	E	G	S	N	
EsNIP3;1	H	P	I	A	L	P	G	S	N	
EsNIP4;1	H	P	V	A	L	P	G	S	N	
EsNIP4;2	H	P	F	A	L	P	G	S	N	
EsNIP4;3	H	P	I	A	L	P	G	S	N	
EsNIP5;1	H	P	I	A	L	P	G	S	N	
EsNIP6;1	H	P	I	A	L	P	S	S	N	
EsNIP7;1	H	P	I	A	V	P	G	S	N	

High conservation of signature sequences or residues was shown in plant PIP proteins. In our study (Table 2), EsPIPs showed a typical NPA motif, a highly conserved ar/R selectivity filter and Froger’s position of F-H-T-R and Q/M-S-A-F-W, these characteristics are correlated with water transport activity (Quigley et al., 2001). In addition to water transport, plant PIPs also could transfer carbon dioxide, hydrogen peroxide, boric acid, and urea (Gaspar et al., 2003; Bienert et al., 2014; Heckwolf et al., 2011). According to the SDP analysis proposed by Hove & Bhave (2011), all EsPIPs had H2O2-type and urea-type SDPs (Table 3, Fig. S2). In addition, all EsPIP1s and EsPIP2;5 had boric acid-type SDPs, and all EsPIP1s had CO2-type SDPs, including two novel types of SDP showed in EsPIP1;3 and EsPIP1;4 which have an M in place of I in SDP2, it also have been found in RcPIPs, JcPIPs and BvPIPs (Zou et al., 2015; Zou et al., 2016; Kong et al., 2017). In addition, EsPIP2;4 owned another novel CO2-type SDPs (V-I-C-A-V-E-W-D-W), with E replaced by D in SDP6. These results showed the conservation of plant PIPs in the transport of urea and hydrogen peroxide (Gaspar et al., 2003; Bienert et al., 2014), and PIP1s not PIP2s are main CO2 and boric acid channels (Heckwolf et al., 2011).

Compared to PIPs, TIPs are more diverse which have a variety of selectivity filters. Two typical NPA motifs were found in all the EsTIPs, and the ar/R filters and Froger’s position were conserved in the EsTIP1s and EsTIP2s classes, but different with other classes. All the EsTIPs showed urea-type SDPs, and most of them had H2O2-type SDPs (except for EsTIP3;1 and EsTIP5;1). EsTIP2;1 had an NH3-type SDPs, as confirmed in Arabidopsis TIP2;1 (Loque et al., 2005). EsTIP3;1 possessed a novel NH3-type SDPs (T-L-G-T-A-S-H-P-A) with F/T replaced by G in SDP3. The NIP subfamily has low intrinsic water permeability and the ability to transport solutes like glycerol and ammonia (Choi & Roberts, 2007). Most EsNIPs held two typical NPA motifs, but some varied at the third residue in the first or second NPA motif. All EsNIPs had urea-type SDPs, EsNIP1;2, EsNIP3;1 and EsNIP5;1 had H2O2-type SDPs. EsNIP5;1, EsNIP6;1 and EsNIP7;1 had boric acid-type SDPs, which have been found in Arabidopsis (Takano et al., 2006). EsNIP1;2 possessed a novel NH3-type SDPs with a substitution of G for A at SDP4. In addition, EsNIP4;1 and EsNIP4;3, which both had the substitution of T for K/L/N/V at SDP2. EsSIPs varied in the third residue of the first NPA motif, with diverse ar/R filters and Froger’s positions. However, the residues were consistent with the corresponding SIP in Arabidopsis. AtSIP1;1 and AtSIP1;2 could transport water in the ER. AtSIP2;1 might act as an ER channel for other small molecules or ions (Ishikawa et al., 2005), and their similarity in these motifs suggests that these EsSIPs may have similar function. These results indicate that the diversity of AQPs in E. salsugineum may have crucial role in response to environmental stress.

Distinct expression profiles of EsAQP genes in various organs

Previous studies have shown that many AQPs show similar expression patterns, suggesting that they may act synergistic in some organs. For instance, PIPs and TIPs are abundant in all organs in many plant species (Quigley et al., 2001; Venkatesh, Yu & Park, 2013; Reuscher et al., 2013; Zou et al., 2015; Yuan et al., 2017). The qRT-PCR results showed that the transcripts of EsAQP genes could be detected in all organs, but their expression levels were diverse (Fig. 6A). Among them, the most abundant transcripts were EsPIPs and a few EsTIPs (EsTIP1;1 and EsTIP1;2), which were consistent with previous studies, especially with Arabidopsis AQP genes (Jang et al., 2004). The high expression of these AQP genes may be related to their effective water channel function that mediates water uptake in plant (Jang et al., 2004; Gomes et al., 2009). Moreover, EsTIP3;1 and EsTIP3;2 were highly expressed in silique specifically. It has been reported that seed-specific TIP3;1 and TIP3;2 play a role in maintaining seed longevity, and as target genes of ABI3 transcription factor which known to be involved in seed desiccation tolerance and seed longevity (Mao & Sun, 2015). It suggested that TIP3s may be involve in cellular osmoregulation and maturation of the vacuolar apparatus to support optimal water uptake and growth of the embryo during seed development and germination (Shivaraj et al., 2017). In general, the transcript level of NIP subfamily is lower than others. However, the EsNIP5;1 was high abundant in all organs, and some of them showed organ specific. For example, EsNIP2;1 and EsNIP3;1 were predominant expression in root, EsNIP4;1, EsNIP4;1 and EsNIP7;1 were predominantly expressed in flower. These may rely on their transport function of diverse substrates (Mitani-Ueno et al., 2011). Strikingly, the SIP1;1 and SIP2;1 exhibited higher expression than many TIPs and NIPs in both E. salsugineum (this study) and Arabidopsis (Alexandersson et al., 2005). Compared with different organs, many AQP genes are mainly expressed in roots and flowers, whereas no AQP isoform is leaf specific in Arabidopsis (Alexandersson et al., 2005). These results were also observed in our investigation. Above all, the parallel expression patterns of AQP genes in different organs between E. salsugineum and Arabidopsis may further indicated their similarity.

Stress responsive AQP genes in E. salsugineum

Environmental stress factors such as salt, drought and low temperature can quickly reduce water transport rates (Javot & Maurel, 2002), thus the maintenance of osmotic potential is a major challenge for plants. Since AQPs are known to be involved in the maintenance of water balance in the plant, we investigated the expression of EsAQP genes at aerial parts of seedlings under various abiotic stresses including salt, drought and cold. In Arabidopsis, most AQP genes are down-regulated upon drought stress in leaves, with the exception of AtPIP1;4 and AtPIP2;5, which are up-regulated (Alexandersson et al., 2005). Besides, the expression analysis of AtPIPs at aerial parts show that only the PIP2;5 was up-regulated by cold treatment, and most of the AtPIP genes were down-regulated by cold stress whereas less-severely modulated by high salinity (Jang et al., 2004). In our data (Fig. 6B), major AQP genes of E. salsugineum were down-regulated expression to drought treatment, however, nine genes (EsPIP2;4, EsPIP2;5, EsTIP1;2, EsTIP2;3, EsTIP3;2, EsNIP1;2, EsNIP4;3, EsNIP5;1 and EsSIP1;2) were up-regulated. Among these, the level of EsTIP3;2 was most significantly increased after drought treatment, which has low abundance in leaf (Fig. 6A). It is suggested that EsTIP3;2 may play a unique role under drought stress. While most of AQP genes were up-regulated under salt stress, it is consistent with those in barley and bamboo (Hove et al., 2015; Sun et al., 2016). Contrary to Arabidopsis, most of AQP genes in E. salsugineum were up-regulated under cold stress. This type of expression pattern has been reported in Sorghum bicolor (Reddy et al., 2015), to improve water transport efficiency and enhance cold tolerance (Li et al., 2008). Moreover, EsPIP1;5 was down-regulated under abiotic stresses but highly abundant in all organs, the EsTIP1;2 and EsNIP5;1 were highly abundant in all organs and up-regulated under various stresses. These AQP genes were induced by external stimuli, and implied to play role in maintaining water homeostasis during environmental stress (Jang et al., 2004).

Conclusions

In our study, a genome-wide information of E. salsugineum AQP gene family was provided. 35 EsAQP s, located in seven chromosomes, were identified and divided into four subfamilies based on phylogenetic analysis, which was also supported by the subfamily-specific gene structure and MEME motifs analysis. Furthermore, functional properties were investigated through the analysis of ar/R filters, Froger’s positions and SDPs, which have potential outputs for the widely function of EsAQPs. Moreover, the expression analysis was performed by qRT-PCR, showing AQP genes were widely involved in E. salsugineum organs development and abiotic stress response, and may have the potentially important roles in E. salsugineum. Our work not only provided a full-scale bioinformation of E. salsugineum AQP genes, but also offered a positive assessment for the underlying candidate EsAQPs in abiotic stress response.

Supplemental Information

Table S1 Primers of EsAQPs used in qRT-PCR

Click here for additional data file.

Table S2 Primers used in construction of vectors. The restriction sites were drawn with horizontal lines

Click here for additional data file.

Table S3 Compared the EsAQPs in this study to existing annotation at Phytozme

Mismatched names highlighted in Yellow.

Click here for additional data file.

Table S4 Predictions of subcellular localization of EsAQP genes in WoLF PSORT

Click here for additional data file.

Figure S1 Putative pI and MW of PIPs, TIPs, NIPs, and SIPs from E. salsugineum

Click here for additional data file.

Figure S2 Specificity determining positions (SDPs) analysis of E.salsugineum AQPs from alignments with putative amino acid sequences of AQPs transporting non-aqua substrates

Multiple alignments were performed using ClustalX. The SDPs are highlighted in yellow, mismatch site highlighted in red and the representative sequences are marked in blue. The Genbank accession numbers: AtPIP1;2 (Q06611), AtPIP2;1 (P43286), AtPIP2;4 (Q9FF53), AtTIP1;1 (P25818), AtTIP1;2 (Q41963), AtTIP1;3 (NP_192056), AtTIP2;1 (Q41951), AtTIP2;3 (Q9FGL2), AtTIP4;1 (O82316), AtTIP5;1 (NP_190328), AtNIP1;2 (Q8LFP7), AtNIP5;1 (NP_192776), AtNIP6;1 (NP_178191), CpNIP1 (CAD67694), GmNOD26 (P08995), HvPIP1;3 (BAA23745), HvPIP1;4 (BAF33068), HvPIP2;1 (BAA23744), NtAQP1 (O24662), NtTIPa (Q9XG70), OsNIP2;1 (Q6Z2T3), TaTIP2;1 (AAS19468), TaTIP2;2 (AAS19469), ZmPIP1;1 (Q41870), ZmPIP1;5 (Q9AR14).

Click here for additional data file.

The authors appreciate those contributors who make the Eutrema salsugineum genome data accessible in public databases.

Additional Information and Declarations

Competing Interests

Author Contributions

Data Availability

The authors declare there are no competing interests.

Weiguo Qian conceived and designed the experiments, performed the experiments, analyzed the data, contributed reagents/materials/analysis tools, prepared figures and/or tables, authored or reviewed drafts of the paper, approved the final draft.

Xiaomin Yang, Jiawen Li and Rui Luo performed the experiments.

Xiufeng Yan conceived and designed the experiments, contributed reagents/materials/analysis tools, approved the final draft.

Qiuying Pang conceived and designed the experiments, contributed reagents/materials/analysis tools, authored or reviewed drafts of the paper, approved the final draft.

The following information was supplied regarding data availability:

The primers used in qRT-PCR are available in Table S1. The specificity determining positions (SDPs) analysis of E.salsugineum AQPs from alignments with putative amino acid sequences of AQPs transporting non-aqua substrates are available in Fig. S2.

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
