# Peer review of "Genome-wide characterization and expression analysis of aquaporins in salt cress (Eutrema salsugineum)"

_PeerJ, doi:10.7717/peerj.7664_

## Round 0.1 · original submission · Major Revisions

I have received three reviews of your resubmitted manuscript. Reviewers agree that the manuscript represents an interesting topic that would potentially be suitable for publication in PeerJ. However, the reviewers also agree that the manuscript is often poorly described and at times superficially understood and implemented, that the Results are in places insufficiently explained, specifically the AQP locations and their physiological function. Furthermore, the discussion is too long and the conclusion is merely descriptive. All reviewers shared the same concerns about the English language, a concern I also share. However, the manuscript has some clear potential to be turned into an interesting and useful contribution that might be suitable for PeerJ. Therefore, I urge the authors to address all criticisms outlined by the reviewers very seriously and thoroughly, one-by-one, by reporting in the rebuttal letter the modified text verbatim, including also the specific line numbers of the changed text, and by explaining how their revised text addresses each criticism raised by the reviewers. Only then can this manuscript be re-considered for potential publication in PeerJ.

In the light of these views I am sorry to say that we cannot proceed further with your paper in its present form. However, if you can see your way to dealing satisfactorily with the various criticisms we would be happy to consider a resubmission of this work.

·

Basic reporting

The paper is interesting and merits publication in PeerJ, but manuscript needs major improvements before being suitable for publication.

The use of english must be improved and need a style revision. In the present form many sentences are confusing or difficult to understand.

Literature references and introduction is Ok.

Article structure, figures and table presentation is Ok.

The hypothesis and the way to adress it is Ok.

Experimental design

Table1 has a major problem. All the localizations are in the plasma membrane or the tonoplast, and only 1 in the chloroplast. It is difficult to believe that Eutrema has no mithocondrial located aquaporins (TIP5;1 is an aquaporin specifically targeted to pollen mitochondria and is probably involved in nitrogen remobilization in Arabidopsis thaliana
Gabriela Soto et al) or in the ER (Plant Cell Environ. 2018 Dec;41(12):2844-2857. doi: 10.1111/pce.13416. Epub 2018 Sep 12.BvCOLD1: A novel aquaporin from sugar beet (Beta vulgaris L.) involved in boron homeostasis and abiotic stress.Porcel R et al) Clearly the algorithm is not working properly. Authors must complete the table with the published localization of the conserved aquaporins in order to give more accuarte information.

Validity of the findings

Findings are interesting, but the presentation of them is very poor and must be improved.

The problem with the present paper are not the experiments itself, but the way they are commented or the conclusions drawn need major improvement.

The discussion is too long and the conclusion is merely descriptive. it is clear that aquaporins are very similar to the ones found in arabidopsis, as expected, but which are the diferences? Do these diferences justify the differential salt tolerance of both species? Authors should redran the discusion and the conclusion and focus in which are the major findings of this papers, specifically, in the differences from arabidopsis, rather than just a lengthy description of the conservation with arabidopsis, very redundant with the results section. Which aquaporins are differentially expressed? Which of them are not conserved, if any? These questions should be clearly adressed and the reader should not look for them among a mess of irrelevant descriptions.

Data is ok.

Additional comments

Please, mind the previous suggestions and the paper will be greatly improved.

Reviewer 2 ·

Basic reporting

The study presents the genome-wide analysis of aquaporin (AQP) genes in salt cress (Eutrema salsugineum). However, the English language of the manuscript should be improved to ensure that your international audience can clearly understand.

Experimental design

In this manuscript, two genome assemblies at scaffold or chromosome level were used in this study. In my opinion, the chromosome-scale assembly is more suitable, however, the authors first identified AQP genes from the scaffold-level assembly and then aligned to the chromosome-scale assembly. Please explain why? Moreover, comprehensive comparison of AQP genes from two assemblies is highly recommended.

Validity of the findings

Several analyses (e.g. gene structure and motif analysis) presented in this study were purely based on the automatic genome annotation. I am afraid it would be misleading. More evidence should be provided.

Additional comments

Other suggestions are as follows:
1. Change “functioned” to “functioning” in line 17.
2. Rewrite lines 24-33.
3. What is the latin name for moss in line 59.
4. Rewrite lines 60-61.
5. Change “stress” to “stresses” in line 79 and 92.
6. Rewrite lines 83-85.
7. Add for after “search” in line 101.
8. Change “Characters” to “Characterization” in line 148.
9. Synteny analysis between salt cress and arabidopsis is highly recommended.

Reviewer 3 ·

Basic reporting

The mauscript is clear and unambiguous.

Experimental design

The original primary research and the research question are fine.

Validity of the findings

The findings in the manuscript seem fine. Its quality wil be much better if the predictions could be tested by experiments

Additional comments

Qian et al have characterizes the AQPs in E.salsugineum, which will favor further studies on AQPs roles played in abiotic stresses. However, I think the manuscript could been improved from the following points:
1. The authors should combine function analysis (AQP function or plant physiological function), but not only the expression data on mRNA level.
2. The primers used should be listed. The PCR efficiency for each primers should be checked in qPCR. The authors did not show the reference gene were stable in stress conditions. Multiple reference gene should be tested and the qPCR should be done according to Vandesompele et al.2002.
3. The authors clarified that all the 35 AQPs were expressed in all organs. How to explain the TIP3s usually specific expression in seeds in other species, for instance in Arabidopsis?
4. Do the authors believe the prediction of the subcellular localization by these two kinds of soft used? The prediction should be tested with AQPs, whose localization has been investigated in other species. And the prediction should be confirmed by experiments.
5. The English should be improved. For instance, the sentence in Line 83 - 85 is not complete.

Minor points:. The authors should explain what is MEME motif.

---

## Round 0.2 · Minor Revisions

The reviewers agree that the manuscript has been largely improved but they still detected problems in the English language. I encourage to perform a English proofreading by a fluent English speaker.

[]

·

Basic reporting

Authors have substantially improved the manuscript, also the english is much better, but please change "conversed" by "conserved"

Experimental design

Meets with the criteria

Validity of the findings

Meets with the criteria

Additional comments

Authors have made a great effort to meet the criteria of the journal

Reviewer 2 ·

Basic reporting

Most concerns have been adressed. However, the English language of the manuscript should be improved before publication.

Experimental design

No comment.

Validity of the findings

No comment.

Additional comments

Most concerns have been adressed. However, the English language of the manuscript should be improved before publication.

---

## Round 0.3 · Minor Revisions

Julin Maloof (one of the Section Editor's for this part of the journal) performed a final evaluation on your submission. He has the following comments for you, which you should address in a final revision:

"This paper ignores the existing annotation for E. Salsugineum. The existing annotation already has numerous aquoporins and TIPs, PIPs, SIPs, NIPs in the annotation. What is the relationship between what is identified here and the existing annotation? Does this improve on the annotation? Other information in the manuscript is OK so this can be published but the relationship between this work and prior work has to be made first. And if all genes "identified" by the HMM pipeline were already annotated as AQP/PIP, etc in the existing annotation then the title needs to be changed from "Identification" to "Characterization". So, I still think the authors should compare their 35 to the existing annotation. Are their TIPs the same as the TIPS in the annotation, etc.

The annotation file "Esalsugineum: Esalsugineum_173_v1.0.annotation_info.txt" is available at phytozyme https://phytozome.jgi.doe.gov/pz/portal.html#!bulk?org=Org_Esalsugineum

---

## Round 0.4 · accepted · Accept

Thank you for the last corrections. The manuscript seems ready for publication.